# Composing games into complex institutions

**Seth Frey** [1,2]*, **Jules Hedges**[3], **Joshua Tan**[4], **Philipp Zahn**[5]

**1** Department of Communication, University of California Davis, Davis, California, United States of America, **2** The Ostrom Workshop in Political Theory and Policy Analysis, Indiana University, Bloomington, Indiana, United States of America, **3** Department of Computer and Information Sciences, University of Strathclyde, Glasgow, United Kingdom, **4** Department of Computer Science, University of Oxford, Oxford, United Kingdom, **5** Institute of Economics, University of St. Gallen, St. Gallen, Switzerland

* sethfrey@ucdavis.edu

**Data Availability Statement:** Our implementation of compositional game theory in the functional language Haskell is available at https://zenodo.org/record/7117797, with implementations of all examples in this text in folder /src/OpenGames/Examples/Governance/. This software allows a

## Abstract

Game theory is used by all behavioral sciences, but its development has long centered around the economic interpretation of equilibrium outcomes in relatively simple games and toy systems. But game theory has another potential use: the high-level design of large game compositions that express complex architectures and represent real-world institutions faithfully. Compositional game theory, grounded in the mathematics underlying programming languages, and introduced here as a general computational framework, increases the parsimony of game representations with abstraction and modularity, accelerates search and design, and helps theorists across disciplines express real-world institutional complexity in well-defined ways. Relative to existing approaches in game theory, compositional game theory is especially promising for solving game systems with long-range dependencies, for comparing large numbers of structurally related games, and for nesting games into the larger logical or strategic flows typical of real world policy or institutional systems.

## Introduction

Game theory, since its development by mathematician von Neumann and economist Morgenstern, has proliferated through the social and biological sciences as a powerful formalism for modeling strategic and cooperative interactions. Economics in particular has applied it to core disciplinary questions, with a keen interest in analytical modeling and the formal properties of game solutions. However, this wildly successful research agenda has obscured promising uses of game theory for which equilibria and other solutions are not the central concern. For instance, other social scientists and engineers have imagined a game theory capable of modeling more integrated multi-stage, hierarchical, or modularized institutions that nest and chain together many mechanisms. The economist Leonid Hurwicz pursued an early conception of institutions as linked systems of games [1], while political scientist Elinor Ostrom introduced the "action situation" framework as an empirically grounded generalization of game theory for structuring ethnographic description [2], and in other work imagined complex institutions as systems of linked action situations [3, 4].

Game theory has long been recognized as a potential tool for the faithful description of realistic social institutions [5, 6]. Calls for this high-fidelity, or "descriptive," game theory have been heard from disciplines as diverse as law [7], international development [8], animal

modular definition of games, and checking different types of Nash equilibria. The library provides a Haskell combinator implementing open games, a battery of examples, and a code generation tool for making the combinator library practical to work with.

**Funding:** Authors SF, JH, JT, and PZ contributed equally to the manuscript. SF acknowledges the support of NSF RCN grant #1917908, "Coordinating and Advancing Analytical Approaches for Policy Design." JT acknowledges members of the Morgenstern workshop at the Mercatus Center, especially Katherine Wright and Ginny Choi, for their feedback and support.

**Competing interests:** The authors have declared that no competing interests exist.

behavior [9], computer science [10, 11], institutional economics [12–15], and sustainability science [16, 17]. Sociologists have articulated generalizations of game theory for the distinct tasks of describing observed institutions [18] and classifying them [19]. And experimentalists have developed many unconventional, interlinked game architectures in pursuit of behavioral insights [20–23]. Approaches for decomposing and recomposing complex institutions in terms of flexible grammars could lead to formal game-theoretic representations for high-level institutional concepts such as distributional justice, polycentricity, and resilience.

These new uses require a scalability, heterogeneity, and overall complexity that existing game design formalisms struggle to capture. Within familiar normal- and extensive-form computational representations (typified by software libraries such as Gambit [24]), each additional player, choice, and stage added to a game contributes to an exponential growth of game outcomes, and a proliferation of equilibria. These threats to game expressiveness highlight the need for a theory of complexes of games that permits modularity, abstraction, and other core principles of engineering, particularly software engineering.

With the introduction of structured programming, and the formal apparatus of modern computer science generally, Edsger Dijkstra and other early researchers abstracted out of machine code to focus on higher-level questions of software architecture [25, 26]. For the same reasons that software engineers have adopted modern computer languages, we offer compositional game theory for designing complex institutions (Fig 1).

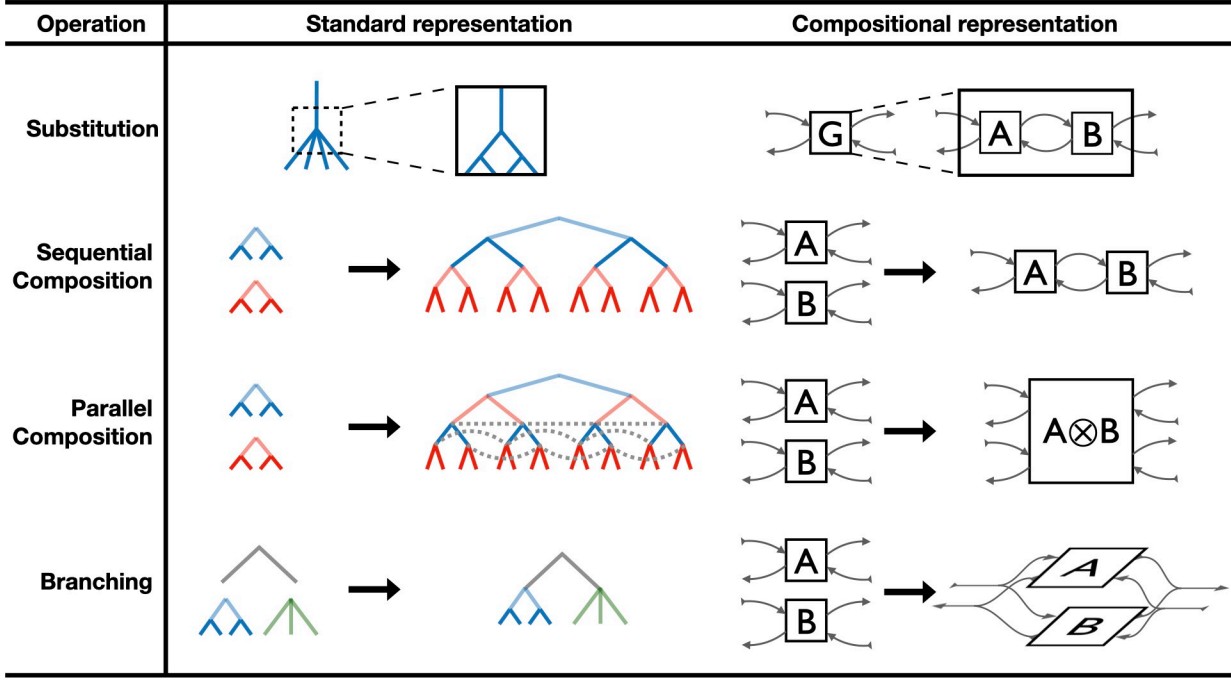

**Fig 1. Transforms illustrating the range of compositional game theory.** We show four illustrative types of abstract game transforms that, when combined, can produce complex institutional forms, toward a high-level, compositional language for linking games into larger systems. Each row shows an extensive form representation of the pattern, followed by a compositional string representation of that pattern. *Substitution* permits modularity and abstraction, the basis of a high-level hierarchical design approach for complex games. *Sequential composition* arranges games in series in a way that abstracts over specific game outcomes, which otherwise grow exponentially in large or repeated games. *Parallel composition* arranges games for simultaneous play in a way that abstracts out of the specifics of complex information sets (dashed lines). *Branching* allows an upstream decision to influence what games are played downstream. It can be seen as providing an XOR choice in contrast to the AND of the other two types of composition. In the cases below, #1 and #2 use substitution, all three use sequential composition, #1 and #2 use parallel composition, and #3 uses branching. With these and other transforms, compositional game theory provides a concise, unified language focused the high-level, architectural dimension of game-theoretic institution design.

The primary aim of this work is to organize prior work on compositional game theory for an audience beyond applied mathematics and theoretical computer science, with a particular focus on interdisciplinary and computational social scientists. Among social scientists, calls for complex games have come from every discipline, but have been most clear and consistent from environmental science and organizational and institutional analysis. Within applied mathematics and mathematical game theory, this work motivates continued formal development by communicating the diversity and importance of its applications.

Compositional game theory articulates traditional game theory in terms of category theory, a branch of mathematics that has been used fruitfully to map software engineering concepts into domains such as quantum computing [27], chemistry [28], and natural language processing [29]. We show, across three cases, how compositional game theory can expand the scope of game theory while supplementing existing ethnographic and other empirical methods. Under the compositional framework, designers can nest games within each other, give players choices between games, and create complex logical flows across games or long-range dependencies within them. Compositional game representations abstract nonessential details of specific games to allow systems of games to be compared, allowing modelers to capture connections between game architectures, and how a game's solutions change with its embedding in different social or policy contexts. In this way, the compositional approach opens several subjects to more practical analysis: large chains of many games, comparisons of structurally similar games, complex logical/strategic flows through games (games of games), and the efficient interactive design of all of the above. It also supports a formal visual string diagram language, and permits designers to quickly prototype new architectures and map proven ones into new contexts. Game complexes are still compatible with existing solution concepts and proof methods, but the theory operates at a more abstract level that focuses modelers on the high-level work of composing and extending them.

To be explicit, this work does not contribute to game theory by offering new equilibria or faster solutions to existing equilibria, and its contributions to the examples we explore below is not to solve them. Construed broadly, classical game theory has no formal limitation that compositional game theory overcomes. And the compositional approach does not solve the problem that large complexes of games may have dozens or hundreds of solutions under familiar solution concepts. But merely introducing compositionality brings attention to the need for solution concepts that are selective enough to aid the analysis of very large games. Compositional game theory gives modelers and designers a framework for exploring and iterating over arbitrarily large games, but it is more than a representational advance. By extending the range of social systems that can practically be expressed game theoretically, through improved designability, extensibility, comparability, and visualization, it is also a source of new questions about the game theoretic study of institutions.

## Compositional game theory

Compositional game theory is a formal framework for composing economic games into larger systems of games [30–32]. It is grounded in category theory, especially the categorical approach to open systems [33, 34]. In technical terms, this approach models systems as morphisms $f{:}X{\rightarrow}Y$ in a symmetric monoidal category where the objects $X$ and $Y$ describe the boundaries of the open system. This is notable because of the connection it reveals between the structures of game theory and software architecture. Classic models of computation such as lambda calculus have been productively modeled using category theory [35]. As well as abstraction and modularity, open games admit a formal graphical representation [36, 37] that is closely related to other formal diagram systems, such as Feynman diagrams [38].

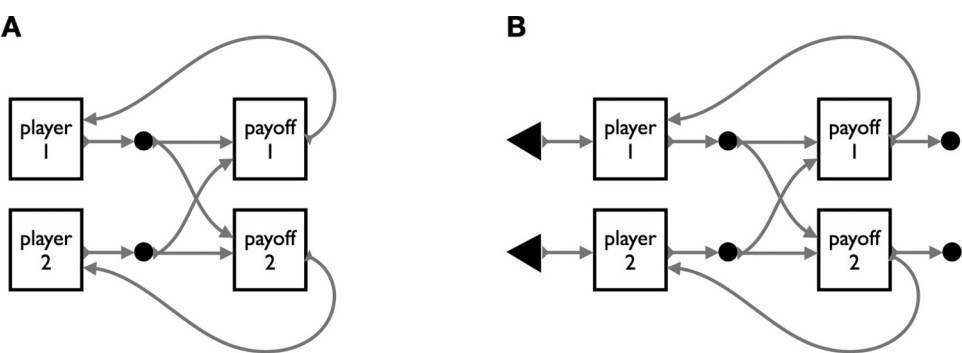

**Fig 2. Closed and open versions of the string diagram of an *n*-choice, two-player game.** Two players emit decisions based on prospective information about (only) their own payoffs. Their decision feeds into the calculation of their own payoff and that of the other player. Time proceeds left to right, with players making decisions that emit payoffs. Arrows feeding backwards in time to players represent player preferences over future events and signify the presence of strategic reasoning. In these diagrams, specific choice sets and payoffs are abstracted away, improving parsimony as games scale. Formal string diagrams of this style map directly to game architectures in the sense that the computational representation of a game could be compiled to or from its diagram. Boxes are general, and can be used to represent players, payoffs, decision nodes, entire games, or any potential target of substitution. **A.** This "closed" version of a game is consistent with conventional game theory. Players are fixed, and results of the game feed back to those players. **B.** The closed game can be opened with the addition of inputs and outputs, represented by incoming or outgoing arrows. In the open version, players are replaced by inputs of player type, enabling flexibility as to how agents are selected to fill the player role, and reuse of the game in different contexts. This version of the game also has open outputs. In addition to feeding payoff information back to the players, it emits them as outputs that could, for example, be used to parameterize a downstream game. We show this in Fig 5, an irrigation social dilemma, which models the steady depletion of a water level variable by making the output of one decision unit the input of the next.

Framed within category theory, a game is a kind of process, following the arrow of time from past to future (Fig 2). And the basic unit of categorical game theory, the open game, generalizes a game so that it can communicate with an external environment through its inputs and outputs, which define its type. Open games connect along their type boundaries to compose into larger open games in such a way that each component becomes a part of the environment of the others. This is directly analogous to how modern software is built by connecting standard components—such as functions, classes, and modules—through well-defined interfaces. In fact, the analogy is direct enough that string diagrams of open games can be directly compiled to software and are subject to formal guarantees, such as the guarantee that any composition of open games will be another well-typed open game.

To give a sense of the expressiveness of our approach, we describe several operations and primitives for composing open games into complexes (Fig 1). First we introduce substitution, in which a placeholder for an open game component can be occupied by any system of games as long as its inputs and outputs are of the right type (Fig 1, Row 1). In the second transform, sequential composition, two games are appended "end-to-end" (Fig 1, Row 2) so that they can be played serially. A challenge overcome by this seemingly simple operation is that familiar approaches, such as game trees, grow unwieldy exponentially as more games are appended. The compositional framework can automatically manage this growth. Another operation we define is parallel composition, in which several games are appended "side-by-side" for simultaneous play (Fig 1, Row 3). Existing representations can capture the complex information sets that come with parallel composition, but as with sequential composition, they become very difficult to manage as the number of composed games increases. A fourth transform we introduce is branching, which permits a game outcome to output not just payoffs, but system parameters and pointers to games and players (Fig 1, Row 4). With branching, the outcome of a game representing the policy design process is not a set of payoffs, but another game representing the

designed policies. These four patterns are not exhaustive, but as a subset of possible patterns, they enable a broad range of game operations, as we show in three examples below.

## Case 1: $CO_2$ certificate markets

Markets for emissions are a prominent tool in the economic fight against climate change. As in the analysis of climate negotiations [39], game theory has played a crucial role in research on these sometimes complex institutions [40].

Institutional arrangements like emission markets are interesting because their many moving parts can interact in unexpected ways. Consider a simplified $CO_2$ certificate market, which involves an allocation stage for initially distributing certificates, a production stage in which players generate $CO_2$, a resale stage for trading partially and over-fulfilled certificates, and a second production stage (Fig 3). A researcher might ask several questions. How should the initial permits be allocated? How should the resale market be structured? What are the distributional effects on producers? How are consumers affected?

These questions hinge on the subtle, long-range interactions between stages of this market. For instance, details about the stage one allocation of permits can have an indirect effect upon the stage three resale market [41–43]. In this system, the allocation and resale stages are strategic (as indicated by the presence of backwards-facing arrows, signifying a decision that depends on prospects about future interdependent outcomes). By contrast the production stages are non-strategic, in the sense that they can be made entirely on the basis of information that doesn't depend on the decisions of others. But once embedded between strategic decisions, production decisions begin to figure into a firm's strategic reasoning. Existing models of

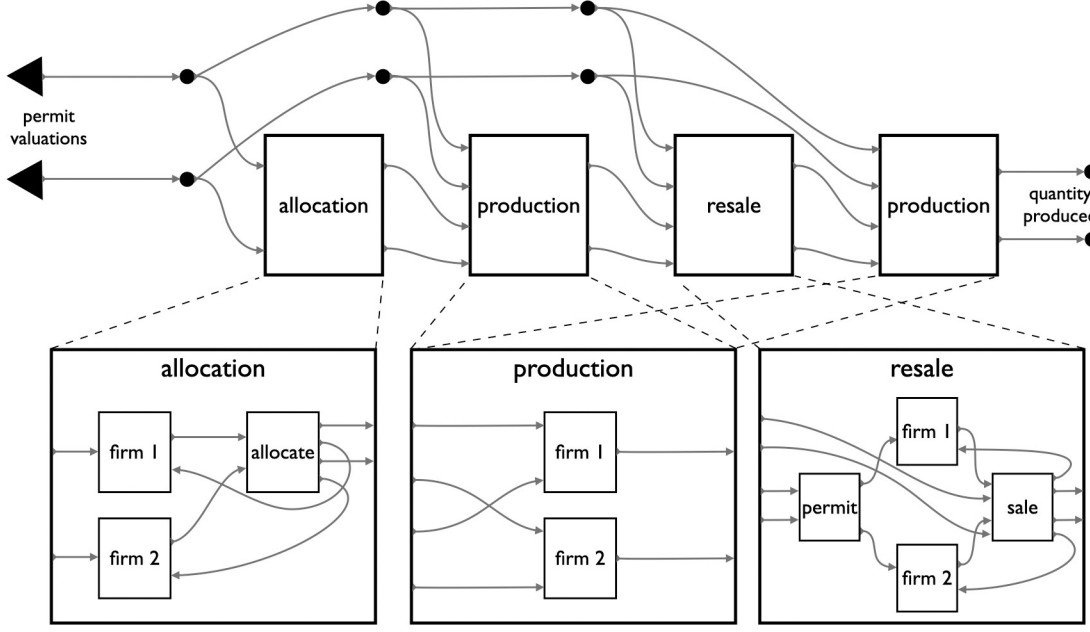

**Fig 3. A four stage $CO_2$ market game.** This multi-stage game proceeds through an initial allocation stage, a production stage, a resale stage, and a second production stage. The first models the primary allocation of $CO_2$ certificates to producers. Producers who received permits then decide how to use them in production. Afterwards, they either have unused permits left or are seeking further permits, and so participate in a resale market that is then followed by a final production phase. Producers operate under incomplete information: they do not know how highly others value their permits. With each stage represented as modules, stages like the production stage can be reused. The explicitly typed incoming (large left-pointing triangles) and outgoing arrows (terminating in circular nodes that represent the game's composability) make this complex of open games itself a game that could be opened and embedded within a larger game.

these markets often focus on one or two stages in isolation, resulting in a collage of models which succeed in analyzing different aspects but fail to provide a global, integrated view of the full process, or how it will play out differently in different contexts.

Compositional game theory offers an alternative modeling approach, one that brings to game theory capabilities for modularity, reuse, abstraction, and other principles of programming. Individual components are modeled with an interface relative to an environment. As with traditional models modelers can zoom in on specific components and analyze them in isolation. They can also substitute parts in the process of modeling. In the compositional approach, the required interfaces constrain the modeling of each stage to ensure that it can communicate with the rest of the model. Modelers thereby gain a theoretical framework for iterating systematically through variants of a large connected system, while being able to freely switch between traditional equilibrium analysis and other methodologies like simulation.

## Case 2: Nepali irrigation monitoring schemes

From fisheries to pastures, forests, and irrigation systems, communities around the world depend upon the successful community management of common-pool resources. But because communities differ greatly from each other, the principles of success can be elusive. This is especially clear in studies such as those by Ostrom and colleagues [44, 45], who compared the collective action institutions of hundreds of small-scale irrigation systems in Nepal.

These works examined institutional diversity, the range of successful approaches to a given collective action problem: when there is asymmetrical access to limited water by upstream farmers, "head-enders" can leave "tail-enders" with insufficient resources. Farmer communities have successfully evolved many different institutions for solving this dilemma, but they are difficult to compare with existing tools. Communities have been observed to rotate water access by season, crop, farmer, or day of the week. They may or may not rely on monitors to enforce local rules. Those that do might pay their monitor fixed fees, fractions of crop yields, or they might allow their monitor to administer and retain penalties. Represented as games, farmer players can use water profligately or equitably, monitors can exert costly work or shirk, and each regime interacts with these choices differently.

With game modeling tools focused on fine-grained institutional features, it can be difficult to see the features that such diverse institutions have in common against the noise of their differences: differing numbers of players, numbers and types of choices, specific payoffs, and other factors. Kimmich, for example, models an irrigation governance system as a network of six adjacent games, to show how incremental changes in one part of the network ripple through to affect the equilibria elsewhere [16].

Using the compositional framework, we developed a simple grammar for capturing institutions in the Nepali irrigation case, to rapidly build and test different observed variants. Under a framework focused on architecture, designers and modelers can iterate efficiently while managing the cascading effects of structural changes (Fig 4). The result represents the range of regimes against the background of their structural similarities (Fig 5). One thing that becomes apparent is that the process of structuring incentives toward greater fairness requires increasing the complexity of the institution and decisions within it. As indicated by the monitor's backwards-facing arrows in Fig 5C and 5D, the variations with greater capacity for fairness are also those with a greater number of interacting strategic decisions.

We developed these variants through an iterative, exploratory process. One conclusion of this comparative modeling exercise was that we cannot engineer a unique Nash equilibrium that is equitable, with all farmers extracting the same amount, without carefully combining several types of incentives. The flexibility and extensibility of the compositional framework

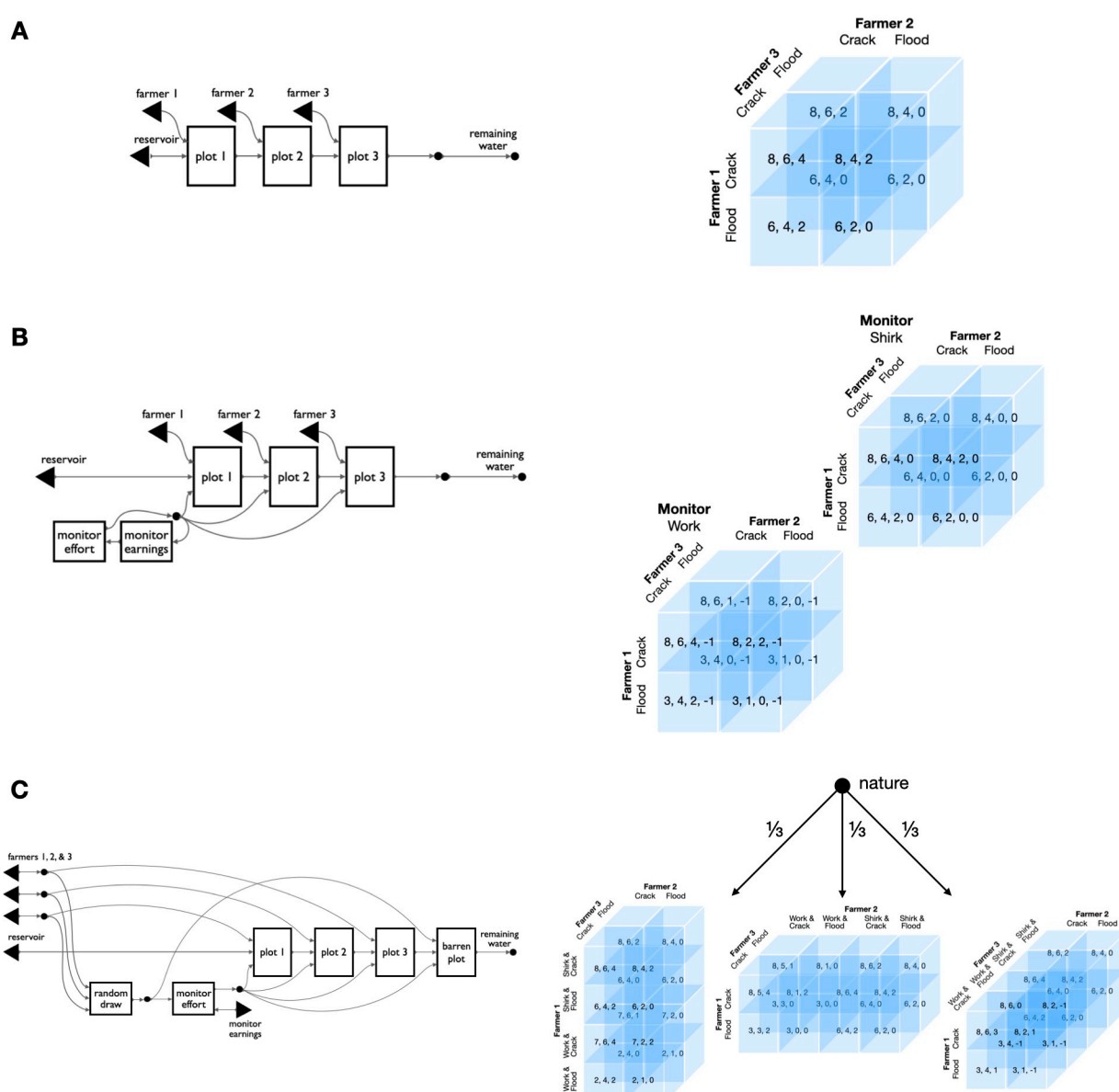

**Fig 4. String diagrams make complex games easier to visualize, and their variants easier to compare.** From an institutional design perspective, adding a player to a game, adding actions to a player, or adjusting game parameters should all be minor changes. But in typical game representations they often result in large and complicated game matrices of many different forms. In open games these same design changes can be implemented with correspondingly minor adjustments. To illustrate the difference, we show the three closely related game variations of Fig 5 next to their normal form representations. Compositional game theory captures this family of irrigation games by abstracting away from payoffs and outcomes to focusing on the structure of each game's general dependencies. By contrast, the normal form representations of these games are too different—visually and structurally—to preserve their family resemblance. A. The simplest variant (from Fig 5B) is the default irrigation system with three farmers and no monitoring. In normal form it corresponds to a 2x2x2 cube, mapping 24 payoffs to 8 outcomes. This cube may not be familiar as a "conventional" game representation. This is because most uses of the normal form are for games of two, not three, players. B. The next variant in Fig 5 adds a fourth player, the external monitor (Fig 5C), by including two boxes for the additional player and appropriate links from those boxes to the base game. In normal form, this same game is a 4-dimensional (2x2x2x2) hypercube, represented here as one cube for each of the outside monitor's actions. Changing the costliness of monitoring effort (here 1 unit) requires changes to 8 cells of the 16 outcomes, and changing the punishment (here 50% of earnings) requires changes to 7 cells. C. The third game from Fig 5 implements random assignment of the monitor role to one of the three farmers (Fig 5D). Although it returns to only three players, its representation in familiar systems is the most complex of the three examples. This game does not have a representation in normal form, but can be represented as an extensive-form game against nature that provides a uniform probability of selecting three different normal form games: a 4x2x2 game, a 2x4x2 game, and a 2x2x4 game, each increasing the choice set of one farmer from 2 to 4 actions. As we emphasize elsewhere, simple structural diagrams are not our key contribution, they are a side effect of the underlying computational representation that compositional game theory makes possible.

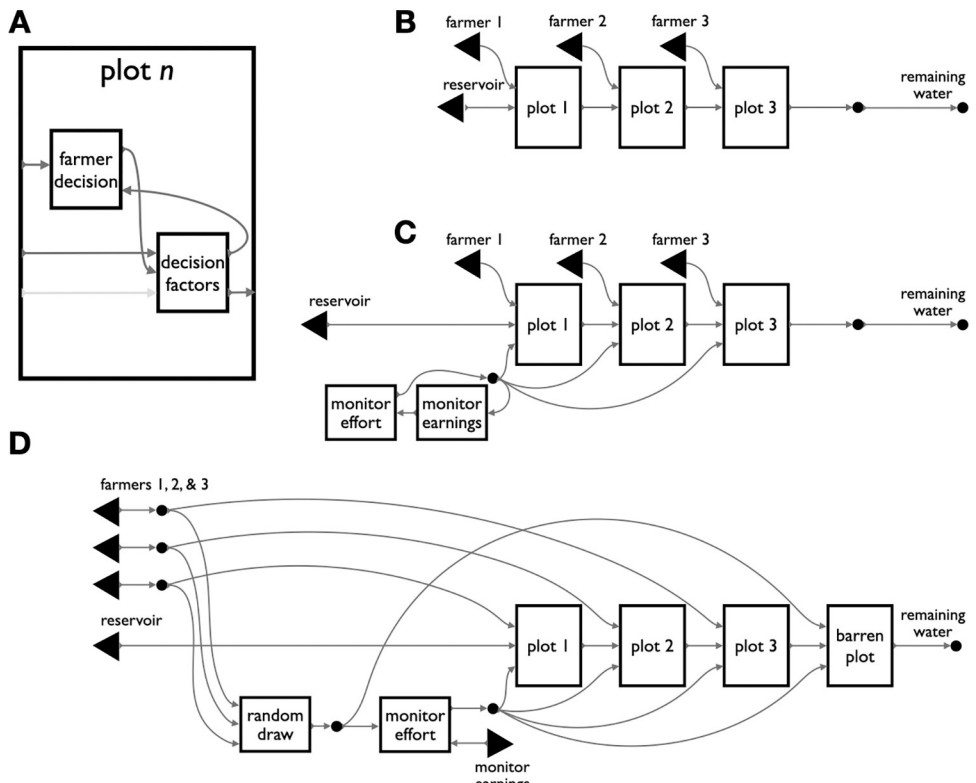

**Fig 5. Several variations of an irrigation institution.** Compositional game theory brings modeling attention to high-level features such as game structure and evolution. In a sustainability application, it can capture the diversity of solutions to the asymmetric social dilemmas typical of rural irrigation systems. The variants here are drawn from a comparative study of water sharing institutions in Nepal. **A.** This module of game structure represents the internals of each plot of the other panels. Farmers decide how much water to extract for their plot on the basis of their incentives, which are calculated from many different inputs. In a direct analogy to function declarations in many programming languages which define the types of the inputs and outputs, this module can be seen as a function, its inputs are a player string, a water level parameter, and an optional penalty parameter, and its single output is an updated water level parameter. This module's reuse in the other games, with different players entered in different ways, and water levels output from prior calls being used as the input to subsequent calls, all reflect the substance of the mapping from software design to institution design that compositional game theory permits. **B.** In the simplest institution, upstream "headender" farmers extract water from a finite reservoir without concern for the water needs of "tailenders." At typically low reservoirs, no water remains for lower plots. **C.** With minor modifications to B, an external monitor earns a fixed rate to enforce sustainable extraction with penalties. This variant allows the monitor's action to influence the farmer's decision by using the optional third input in the plot module (panel A). **D.** In this sophisticated variant, the farmers rotate through the monitor role, who is incentivized to work honestly with access rights to a fourth field that only receives sufficient water when all upstream farmers are compliant. Compositional game theory facilitates high-level comparison across cases, and iteration through them.

permitted us to come to this conclusion through a rapid and interactive exploratory design process. We started with the game of Fig 5B and ideas for possible extensions (including external monitoring, punishment, and peer monitoring with several kinds of incentives, all appearing in different combinations in the other panels). We then worked through the list of additional mechanisms, scanning single parameters for how they changed the base game's equilibria (S1 File). Through this process we found that no single mechanism could succeed in stabilizing an outcome of equitable cooperation (and that defection by the head-ender is the most common pure strategy equilibrium). Of course, this conclusion holds only for a single iteration of a scenario that, for farmers, repeats every season. With serial composition, the scenario can be extended through time, and represent arbitrary structural complexity.

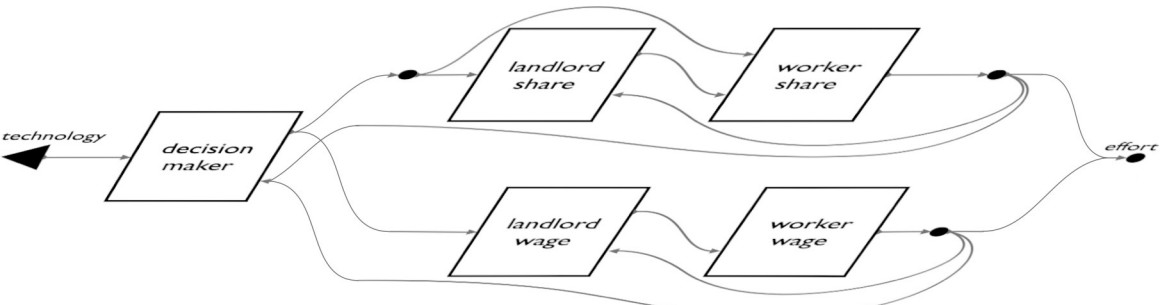

**Fig 6. A decision maker uses technological information to select between alternate policy designs.** In compositional game theory, a player's actions include choices between games. We use this feature to extend Hurwicz's 1996 model of the policy design process. A modular decision maker component—which could be filled in with a single decision maker or collective process such as voting—computes a preference between different policy approaches to a principal-agent problem, each an open game. The policy chosen will decide whether the agent earns a fixed wage or piece rate. The two regimes lie in different planes because they are mutually exclusive. The decision maker's preference between them is informed by the agent's prospective effort in each regime, and also by incoming information about the outside technological environment. For example, technology may improve the observability of worker effort, which may improve the performance of wages relative to the piece rate. With the regime selected, a landlord and worker play out the selected institution, with workers emitting a final effort.

### Case 3: Policy development in the Hurwicz model of institutions

Mechanism design is the area of economics, and social science generally, most concerned with the formal design and engineering of institutional processes. It has been especially successful in simple or narrowly defined applications, and is increasingly common in political science, for models of the policy design process. For instance, a theme that has occupied decades of interest in political science and political economy is that the process by which a policy is developed can have a dramatic effect on its form [12, 46–48]. This phenomenon requires integrated models of the design, adoption, and functioning of a policy.

With an eye to this problem, Leonid Hurwicz, one of the pioneers of mechanism design, introduced a definition of institutions in terms of families of game forms, and applied the construct to model policy processes [1]. Hurwicz imagines a classical principal-agent scenario, in which the incentives facing a landlord and sharecropper are determined by their land tenure arrangement (sharecropping, wage-labor, renting, etc.), which is itself determined by two earlier game stages that establish the parameters of the final game. In developing the formalism, Hurwicz proposes abstracting payoffs out of the process model, making payoffs just one kind of game outcome, and including inputs that represent the game's external environment (the "open" in "open games"). Compositional game theory leverages advances in theoretical computer science to offer an abstract, modular basis for Hurwicz's approach. Within a compositional approach, a designer can specify a policy selection process that leaves the specific policy as a black box. Any set of policies with matching type can be substituted in as a module.

We implement a compositional rendition of the Hurwicz landlord scenario (Fig 6), in which an employing principal and employed agent interact under a land tenure arrangement selected by an outside decision maker. By overcoming the representational limits of existing game forms, compositional game theory consummates Hurwicz's vision for a generalized game theoretic approach to policy processes. In the process it extends the range of institutions that can be expressed as complexes of games.

## Conclusion

Our contribution is to introduce computational social scientists to a theoretical framework for high-level game architecture, grounded in the mathematics of software engineering.

Compositional game theory introduces modularity, abstraction, and expressive power to game theoretic institutional design. It is a principled extension of game modeling to systems with complex interlinkages and multiple levels, toward a rigorous computational representation of real-world institutions. The need for compositional game theory rests in part on the growing need for a high-level game theory interested in richer and more facile game representations, as well as a descriptive game theory focused less on the formal solutions and solvability of games and more upon expressing the structural variety observable in case, ethnographic, and historical work in all disciplines.

We offer a lexicon of game transforms and design patterns that illustrate the compositional approach to institution design in areas of economic mechanism design, sustainable resource management, and policy design. By formally extending game theory to permit compositionality, we meet a need that has been expressed across the behavioral sciences: a design framework for complex systems of games.

## Supporting information

**S1 File.**
(PDF)

## Acknowledgments

JT acknowledges members of the Morgenstern workshop at the Mercatus Center, especially Katherine Wright and Ginny Choi, for their feedback and support.

## Author Contributions

**Conceptualization:** Seth Frey, Jules Hedges, Joshua Tan, Philipp Zahn.

**Data curation:** Jules Hedges, Philipp Zahn.

**Formal analysis:** Jules Hedges, Philipp Zahn.

**Funding acquisition:** Seth Frey, Joshua Tan.

**Investigation:** Seth Frey, Jules Hedges, Joshua Tan, Philipp Zahn.

**Methodology:** Seth Frey, Jules Hedges, Philipp Zahn.

**Project administration:** Seth Frey, Jules Hedges, Joshua Tan, Philipp Zahn.

**Resources:** Joshua Tan.

**Software:** Jules Hedges, Philipp Zahn.

**Supervision:** Seth Frey, Jules Hedges, Philipp Zahn.

**Validation:** Seth Frey, Jules Hedges, Philipp Zahn.

**Visualization:** Seth Frey, Jules Hedges, Philipp Zahn.

**Writing – original draft:** Seth Frey, Jules Hedges, Joshua Tan, Philipp Zahn.

**Writing – review & editing:** Seth Frey, Jules Hedges, Joshua Tan, Philipp Zahn.

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
