## [Decision Letter · Decision Letter 0]

27 Jun 2022

PONE-D-22-13128 Composing games into complex institutions PLOS ONE

Dear Dr. Frey,

Thank you for submitting your manuscript to PLOS ONE. After careful consideration, we feel that it has merit but does not fully meet PLOS ONE’s publication criteria as it currently stands. Therefore, we invite you to submit a revised version of the manuscript that addresses the points raised during the review process.

We look forward to receiving your revised manuscript.

Kind regards,

Ricardo Martinez-Garcia

Academic Editor

PLOS ONE

Journal Requirements:

2. Please expand the acronym “NSF” (as indicated in your financial disclosure) so that it states the name of your funders in full.

 "Authors SF, JH, JT, and PZ contributed equally to the manuscript. SF acknowledges the support of NSF RCN grant #1917908, “Coordinating and Advancing Analytical Approaches for Policy Design.” JT acknowledges members of the Morgenstern workshop at the Mercatus Center, especially Katherine Wright and Ginny Choi, for their feedback and support. " 

Reviewers' comments:

Reviewer's Responses to Questions

**Comments to the Author**

1. Is the manuscript technically sound, and do the data support the conclusions?

Reviewer #1: Yes

Reviewer #2: Yes

2. Has the statistical analysis been performed appropriately and rigorously? 

Reviewer #1: N/A

Reviewer #2: N/A

3. Have the authors made all data underlying the findings in their manuscript fully available?

Reviewer #1: Yes

Reviewer #2: Yes

4. Is the manuscript presented in an intelligible fashion and written in standard English?

Reviewer #1: Yes

Reviewer #2: Yes

5. Review Comments to the Author

Reviewer #1: This paper provides an overview of compositional game theory and illustrates its applications using three case studies. The authors argue this theoretical framework, by allowing for modularity and abstraction, enables its users to build game-theoretic representations of more complex institutions than would be possible with traditional game theory.

The paper is generally well written. It is intended for an “audience beyond applied mathematics and theoretical computer science, with a particular focus on interdisciplinary and computational social scientists” (p.9). I believe this work would be of interest to not only these communities but also to environmental scientists and organizational economists, among others, as exemplified by the case studies.

My remarks are primarily about the scope and framing of the paper.

Major comments:

- There appears to be a slight discrepancy between the abstract/conclusion and the introduction as to what the main contribution of the paper is. The abstract and conclusion claim that the paper contributes the theoretical framework of compositional game theory (e.g., “Our contribution, compositional game theory, permits another approach of equally general appeal…”, p.7), whereas the introduction states that "[t]he primary aim of this work is to organize prior work on compositional game theory…” (p.9).

I believe the latter is a more accurate description of this paper. The authors have in fact made significant contributions to developing the theory of compositional game theory (e.g., refs 30-32, which include two authors of the current paper), but if I understand correctly, these contributions, including the mathematical formalism, are already published. It seems that the current paper instead reviews the key building blocks of this framework (Figures 1 and 2) followed by concrete applications to three case studies (Figures 3-5) and their implementations (Supplementary Materials)—which are certainly worthwhile contributions, particularly for practitioners and applied researchers interested in using the framework. I would suggest that the authors revise the framing along these lines.

- I find the introduction somewhat abstract. In particular, it is clear that there is interest in a compositional approach to game theory, but it is less clear when classical game theory fails.

The software tutorial (Supplementary Materials, https://github.com/philipp-zahn/open-games-hs/blob/master/Tutorial/TUTORIAL.md) reads “NOTE: In its current form, compositional game theory is essentially a programmable reincarnation of classical game theory. The models you can generate with it can also be generated with classical game theory. What is different is how one approaches that modelling task: it becomes a programming task.” If this is in fact the main difference, I suggest that the authors highlight it explicitly in the main text. And if not, could the authors provide concrete examples of what compositional game theory can do but traditional game theory cannot? (I see attempts at this in the case studies, e.g., “Existing models often focus on one or two stages in isolation…” (p.12), but it’s unclear whether these are fundamental limitations of classical game theory or not.)

Minor comments:

- Case 2 ends with a paragraph claiming that the authors “[could not] engineer a unique Nash equilibrium that is equitable, with all farmers extracting the same amount, without carefully combining several types of incentives” (p.14). Although I am inclined to believe this claim, I do not see any results/figures supporting it. I would suggest the authors provide at least a short verbal explanation for why this is the case or, preferably, some supplementary data showing how they arrived at this conclusion.

- "In compositional game theory, a player's actions include choices between games" (Figure 5, p.15). This is a neat feature of the compositional approach with applications to many other scenarios. Perhaps the authors could highlight it in the main text (in addition to the figure caption)?

Reviewer #2: Overall Comments

In this paper, the authors present the framework of open games for formally representing complex game- theoretic interactions and provide three case-study examples illustrating the range of social-ecological systems and social institutions to which this framework can be applied. The paper starts with a short summary of the idea of open games and showing examples of how baseline games can be extended to describe complex institutions by composing games through methods including substitution, sequential and parallel composition, and branching, explaining how the formal string diagram representation of open games allows researchers to provide a compact representation of complex game architectures. By showing the examples of carbon certificate markets, irrigation monitoring schemes, and a principal agent model, the authors demonstrate how serial composition, branching, and careful examination of game architecture can be used to design and evaluate the role of institutions in the actions of game-theoretic agents.

Overall, I think that this is a useful paper that can help to advertise and communicate the open games framework as a useful tool for studying strategic dynamics in complex social systems. While compositional game theory has been studied in previous papers by the authors taking the approach of applied mathematics or theoretical computer science, this paper will serve as a useful introduction to practitioners in the social sciences who are interested in using systematic approaches to design and explore institutional structure in game theory. In addition, I think that this paper may be useful for applied mathematicians and others interested in the mathematical aspects of open games, as it can provide a tangible motivation for the conceptual framework before embarking on learning the mathematical details. The paper is also well-written overall and presents a useful illustration of the emerging field of compositional game theory, so I believe that this paper is appropriate for publication in PLoS One.

However, I do think that the paper may benefit by including a bit more detail on the case study examples and the reasoning behind the form of string diagrams representing each model. I think that this will help further communicate the ideas of the paper for its intended audience. I also have a few minor comments regarding other aspects of the paper. Addressing some of the these points in a revised draft may help the final version of the paper more accessible to a wide audience of researchers interested in using this approach in applications.

Major Comment

In general, I believe that adding some additional exposition on the construction of the string diagrams for each example open game may be helpful for increasing the accessibility of this paper and the chance that readers may try to follow up on this paper by implementing compositional descriptions to model institutions relevant to their own research interest. Below, I have included a list of points that may be helpful to address in a revised version of the paper. Many of these points may only be an issue due to my own confusion, but these points may be helpful for increasing the readability for a broad audience.

• What do the circles and backwards triangles represent in the string diagram? From context, it seems like the triangles correspond to external inputs to games like institutions and players, and

that the circles are used to represent when a single quantity serves as an input to two or more component games. While it may be feasible for readers to glean these aspects from context clues, I think it may be helpful to provide more description of general features of the string diagrams in the text to clarify how to read or construct representations for open games.

• I think it may be helpful to explicitly mention places where the game compositions are used in the case study examples, such as the use of sequential composition in the three case studies and the use of branching in the Hurwicz principal agent model. This can help tie together the big picture idea of composing games presented in Figure 1 with the examples of designing games presented in the case studies.

• In the irrigation model, is there a reason why the players are only represented externally to the module of game structure taking place in plot n (as represented in Figure 4, panel a)? In particular, should the representation of the baseline actions always look like a version of the string diagram characterizing players and payoffs presented in Figure 2 (when modified for the appropriate numbers of players and actions)? Is the representation with players shown with a backward triangle in the irrigation model used because players can serve as both farmers and institutional monitors in different components of the open game? I feel that addressing some of these assumptions or justifications may be helpful for readers to understand why the example string diagrams take the form shown in the figures, and may help for interested readers to construct their own string diagrams for open games related to their own research.

• In Figure 5, do the squares for landlord/work share or wage correspond to a zoomed out representation of a baseline game scenario with a players action and payoff (as considered in Figure 2)? If so, does such a representation correspond to the substitution operation presented in Figure 1? If the answer to either of these is yes, it is probably worth explaining that in the text. If the construction of this diagram depends on more subtle aspects of the modularity of representations of open games, I think it may be also be useful to mention this in order to highlight the rationale behind the construction of the string diagram.

• How does the use of backward arrows representing preferences work in the different examples and composition rules? For example, what does a backward arrow mean for the different component games represented in Figure 1? In addition, when the farmers are represented by backwards triangles outside of the plot diagrams in Figure 4, should there be backward arrows running from the plot modules back to the triangles representing farmers? If not, why does the player represented by a triangle not have backward arrows corresponding to preferences over the actions in the plots, while players in Figure 2 have preferences over the payoffs? Clarifying this point may help readers in figuring out how they would design their own string diagrams.

• Overall, I believe it would be helpful to include more text justifying the representation of different game architectures. I think this will help in making clear that the string diagrams are a formal method for representing open games, rather than more of an ad hoc approach for representing game interactions like informal flow charts. In addition, this may facilitate follow-up research in which readers can apply the approach of this paper to design their own game architectures and model different institutional incentives with game-theoretic modeling.

Minors Comments

Here are a few additional minor comments which may be helpful to address in the revised manuscript.

• I am slightly confused by the statement in the first paragraph of the introduction that “this wildly successful research agenda obscured promising used of game theory for which equilibria are not the central concern”. There are large fields of research on non-equilibrium dynamics in game theory, including evolutionary game theory and models of learning in games. In addition, my understanding is that the open games framework is a clean way of representing more complex game interactions, but that the games that the framework aims to represent could be studied from an equilibrium framework or using other approaches like dynamical simulation of strategies. Given these factors, I think that the quoted statement in the introduction may be an unnecessary criticism of the existing state of game theory, and perhaps recentering the focus of the introduction on the representation of complex interactions may improve the introduction.

• Is one benefit of the string diagram representation for open games that one can abstract away nonessential details from the specific game interactions, allowing to describe equivalence classes of game-theoretic interactions and institutions that impact strategic behavior? If so, this is probably useful to mention, as it may allow modelers to see connections between game architectures that have been stated in particular applied contexts but should actually have analogous equilibrium and dynamical behaviors.

6. PLOS authors have the option to publish the peer review history of their article (what does this mean?). If published, this will include your full peer review and any attached files.

Reviewer #1: No

Reviewer #2: No

---

## [Editor Report · Decision Letter 1]

2 Nov 2022

PONE-D-22-13128R1

Composing games into complex institutions

PLOS ONE

Dear Dr. Frey,

Thank you for submitting your manuscript to PLOS ONE. After careful consideration, we have decided that your manuscript does not meet our criteria for publication and must therefore be rejected.

After careful consideration of the manuscript, we have concluded that it does not fully adheres to PLOS One publication criteria. Specifically, to be considered as a Methods manuscript, a submission must meet the criteria of Utility and Validation (https://journals.plos.org/plosone/s/submission-guidelines#loc-methods-software-databases-and-tools). To meet the validation criterium, your manuscript is missing a deeper preliminary analysis of the three cases considered. 

I am sorry that we cannot be more positive on this occasion but hope that you appreciate the reasons for this decision.

Kind regards,

Ricardo Martinez-Garcia

Academic Editor

PLOS ONE
---

## [Author Response · Author response to Decision Letter 1]

12 Jan 2023

Our response to these comments is in the uploaded response to reviewers file.

---

## [Decision Letter · Decision Letter 2]

8 Mar 2023

Composing games into complex institutions

PONE-D-22-13128R2

Dear Dr. Frey,

We’re pleased to inform you that your manuscript has been judged scientifically suitable for publication and will be formally accepted for publication once it meets all outstanding technical requirements.

Kind regards,

Peter Edwards

Academic Editor

PLOS ONE

Additional Editor Comments (optional):

As a relative outsider to this field, I am of the belief that this manuscript does satisfy the requirements of PLoS One. I have no ability to comment on the technical aspects of this manuscript, and am relying on the expertise of the reviewers. Both reviewers are satisfied with the manuscript; I would, however, encourage the authors to address the very minor comments the reviewers have made.

Reviewers' comments:

Reviewer's Responses to Questions

**Comments to the Author**

1. If the authors have adequately addressed your comments raised in a previous round of review and you feel that this manuscript is now acceptable for publication, you may indicate that here to bypass the “Comments to the Author” section, enter your conflict of interest statement in the “Confidential to Editor” section, and submit your "Accept" recommendation.

Reviewer #1: (No Response)

Reviewer #2: All comments have been addressed

2. Is the manuscript technically sound, and do the data support the conclusions?

Reviewer #1: Yes

Reviewer #2: Yes

3. Has the statistical analysis been performed appropriately and rigorously? 

Reviewer #1: N/A

Reviewer #2: N/A

4. Have the authors made all data underlying the findings in their manuscript fully available?

Reviewer #1: Yes

Reviewer #2: Yes

5. Is the manuscript presented in an intelligible fashion and written in standard English?

Reviewer #1: Yes

Reviewer #2: Yes

6. Review Comments to the Author

Reviewer #1: I am mostly happy with how the authors addressed my earlier comments. The revised manuscript does a better job of clarifying what the paper does / does not and what the compositional game theory does / does not. Reviewer #2 made important comments about the case studies, and, in addressing them, the authors have improved the clarity of the paper.

I only have a few minor comments:

- Abstract: In the conclusion, the authors now write that “[their] contribution is to introduce computational social scientists to a theoretical framework for high-level game architecture,…”. This has not been reflected in the abstract, which still reads as if the paper introduces compositional game theory (“Our contribution, compositional game theory, permits another approach…”). Please correct this.

- Case 1: “Existing models often focus on one or two stages in isolation, resulting in a collage of models which succeed in analyzing different aspects but fail to provide a global, integrate view of the full process, or how it will play out differently in different contexts.” This is a rather sweeping statement. Plenty of models consider how a process “play[s] out differently in different contexts” — what specific contexts are the authors thinking of? Please add references to contextualize and back up this claim.

- Case 3: “Mechanism design is the area of economics most concerned with design.” Design of what? Incentive mechanisms? Institutions?

Reviewer #2: Thank you to the authors for addressing all of the comments from the previous referee reports. All of the comments that I raised were addressed to my satisfaction, and I think that the updated version of the manuscript will serve as a nice contribution to the literature and will be of interest to an interdisciplinary audience of readers interested in both complex systems and game theory. In particular, I think that the revised version of the manuscript clearly presents how open games serve as a computational framework for understand complex game-theoretic scenarios, and the authors do a good job to emphasize differences from classical approaches in game theory. For these reasons, I am happy to recommend this paper for publication in PLoS One.

7. PLOS authors have the option to publish the peer review history of their article (what does this mean?). If published, this will include your full peer review and any attached files.

Reviewer #1: No

Reviewer #2: No

---

## [Editor Report · Acceptance letter]

15 Mar 2023

PONE-D-22-13128R2 

Composing games into complex institutions  

Dear Dr. Frey:

I'm pleased to inform you that your manuscript has been deemed suitable for publication in PLOS ONE. Congratulations! Your manuscript is now with our production department. 

Kind regards, 

on behalf of

Dr. Peter Edwards 

Academic Editor

PLOS ONE